# Hemodynamic Effects of Alpha-Tropomyosin Mutations Associated with Inherited Cardiomyopathies: Multiscale Simulation

**Fyodor Syomin [1,2,\*]** **, Albina Khabibullina [3], Anna Osepyan [1,3] and Andrey Tsaturyan [1]**

1   Institute of Mechanics, Lomonosov Moscow State University, 119192 Moscow, Russia
2   Peoples' Friendship University of Russia (RUDN University), 117198 Moscow, Russia
3   Mathematics and Mechanics Department, Lomonosov Moscow State University, 119991 Moscow, Russia
\*   Correspondence: f.syomin@imec.msu.ru

**Abstract:** The effects of two cardiomyopathy-associated mutations in regulatory sarcomere protein tropomyosin (Tpm) on heart function were studied with a new multiscale model of the cardiovascular system (CVS). They were a Tpm mutation, Ile284Val, associated with hypertrophic cardiomyopathy (HCM), and an Asp230Asn one associated with dilated cardiomyopathy (DCM). When the molecular and cell-level changes in the $Ca^{2+}$ regulation of cardiac muscle caused by these mutations were introduced into the myocardial model of the left ventricle (LV) while the LV shape remained the same as in the model of the normal heart, the cardiac output and arterial blood pressure reduced. Simulations of LV hypertrophy in the case of the Ile284Val mutation and LV dilatation in the case of the Asp230Asn mutation demonstrated that the LV remodeling partially recovered the stroke volume and arterial blood pressure, confirming that both hypertrophy and dilatation help to preserve the LV function. The possible effects of changes in passive myocardial stiffness in the model according to data reported for HCM and DCM hearts were also simulated. The results of the simulations showed that the end-systolic pressure–volume relation that is often used to characterize heart contractility strongly depends on heart geometry and cannot be used as a characteristic of myocardial contractility.

**Keywords:** mathematical modeling; cardiac mechanics; multiscale simulation; cardiomyopathies; left ventricle remodeling

## 1. Introduction

The inherited cardiac diseases, hypertrophic (HCM) and dilated (DCM) cardiomyopathies, can be caused by mutations in genes encoding sarcomere proteins expressed in heart muscle. At least 31 mutations in the TPM1 gene encoding regulatory protein tropomyosin (Tpm) are associated with HCM, DCM, or, more rear, left ventricular non-compaction [1–3]. HCM is characterized by a thickening of the left ventricular (LV) wall occurring in the absence of other diseases. This remodeling often results in a decrease in the volume of LV cavity and obstruction of the LV outflow tract. DCM is characterized by an increased volume of the LV cavity and a reduced ejection fraction in the absence of coronary artery diseases. The remodeling of the LV geometry is believed to play an adaptive role in the protection of the heart function despite impaired myocardial mechanics [4]. Changes in passive mechanical properties of LV myocardium upon HCM and DCM were also found [5–7].

Tpm is a coiled-coil dimer of parallel $\alpha$-helices that serves as a gatekeeper in $Ca^{2+}$ regulation of the actin–myosin interaction in sarcomeres of striated muscles. The Tpm molecules bind to each other in a head-to-tail manner and form a continuous strand located in a helical groove on the surface of an actin filament. The strand controls the availability of actin sites for the binding of motor domains of myosin

molecules—myosin heads. Another regulatory protein, troponin (Tn), binds Tpm in a 1:1 stoichiometry and forms, together with the fibrillary actin and Tpm, a regulated thin filament. Tn controls Tpm movement with respect to the axis of an actin filament in a $Ca^{2+}$-dependent manner [8,9]. The regulation of muscle contraction is highly cooperative: relatively small changes in intracellular $Ca^{2+}$ concentration cause large changes in force and actin–myosin ATPase rate. Modeling [10–12] suggests that the local movements of a stiff Tpm strand (caused by $Ca^{2+}$ binding to Tn or myosin binding to actin) are transmitted to neighbor parts of the strand, providing high cooperativity.

Tpm mutation Asp230Asn is associated with DCM, while the Ile284Val one is associated with HCM. Both significantly change the $Ca^{2+}$ regulation of the myosin–actin interaction measured in the in vitro motility assay or experiments with single cardiomyocytes [13–15]. These molecular and cellular level changes are believed to underlie the impairment of the heart function. Two questions remain unanswered: (1) how the changes in the actin–myosin interaction at the molecular and cellular levels caused by the Tpm mutations affect the pumping function of the heart and the LV particularly; and (2) how does the remodeling of the LV wall associated with the cardiomyopathies change the heart function? To address these questions, we performed a computer simulation of the heart mechanics using a recently developed multiscale LV model [16] incorporated into a simple lumped parameter model of circulation [17]. A model of myocardial mechanics used in the multiscale model was described previously [18]. It describes all major mechanical features of cardiac muscle including the force–velocity and stiffness–velocity relations, tension responses to step-like and ramp changes in muscle length, high cooperativity of $Ca^{2+}$-force relation, and its length-dependence, etc. Here, we used available experimental data to modify model parameters to account for the changes in the $Ca^{2+}$ regulation of myocardial mechanics at the molecular and cellular levels caused by the cardiomyopathy-associated Tpm mutations. Then we simulated the effect of these changes on the movement of the LV wall during heartbeats. We also simulated the effects of remodeling of the left ventricle—hypertrophy and stenosis of the outflow tract for the Ile284Val mutation and the dilatation for the Asp230Asn one—on the calculated cardiac output. The effects of changes in the passive myocardial stiffness associated with HCM and DCM [5–7] were also estimated.

## 2. Materials and Methods

### 2.1. Cardiac Muscle Mechanics and Regulation

A model of cardiac muscle mechanics was described in detail previously [18]. The myocardium was treated as an anisotropic incompressible material with passive elastic and active stress components. Passive elastic stress was a sum of an isotropic hyper-elastic part and a part caused by the tension of titin filaments in sarcomeres. The overall passive stiffness was highly non-linear and anisotropic. The active tension was essentially one-dimensional acting along the axis of muscle fibers. It depended on two molecular variables that characterize the actin–myosin interaction: the fraction of myosin heads bound to actin $n$, and their ensemble averaged distortion $\delta$. These cell-level variables were defined by a system of kinetic ordinary differential equations (ODE) specifying the interactions of contractile and regulatory proteins. Regulation of the contraction was determined by kinetic variables $A_1$ and $A_2$, which represent the fractions of available myosin binding sites on actin in the overlap zone and outside this zone, respectively. The kinetic equations for the variables accounted for the $Ca^{2+}$ regulation of the thin filaments. The balance equation for $Ca^{2+}$ concentration in cytoplasm included the terms of $Ca^{2+}$ influx (set as a given function of time), $Ca^{2+}$ binding to troponin and to other cytoplasmic proteins, and $Ca^{2+}$ uptake from cytoplasm. Variation of the average micro-distortion depended on the sliding velocity of the myosin and actin filaments, thus being defined by the macroscopic strain rate tensor. The full set of equations describing the passive and active stress components in terms of continuum mechanics and the kinetic equations for the interaction of contractile and regulatory proteins and $Ca^{2+}$ dynamics are given in [18] and Appendix A. For the values of the model parameters, see Supplementary Materials Table S1.

### 2.2. Geometry of the Left Ventricle

The LV was approximated by a body of rotation with a shape and distribution of the fiber orientation similar to those observed in human hearts. Despite the axial symmetry of the LV model, all three strain components—radial, axial and angular—were present, so that the ventricle was able to expand, contract, and twist during a heartbeat cycle. The fiber angle with respect to a plane perpendicular to the axis of symmetry changed linearly from $+80°$ at the endocardium to $-55°$ at the epicardium. To describe an increase in ventricular stiffness near its base and the peculiarities of ventricular anatomy near the apex, an additional anisotropic elastic stress of circumferential collagen fibers was added to the stress tensor in the base, and an isotropic elastic stress term depended on the number of bound myosin heads was added in the apex to account for the fiber disorder. The LV model used here was described in detail previously [16,17] (see also Appendix A and Supplementary Materials Table S2).

### 2.3. Model of Circulation

A lumped parameter model of systemic and pulmonary circulations that included the LV ventricle model treated the atria and the right ventricle as viscoelastic reservoirs with non-linear passive and time-varied active stiffness and viscosity [17]. Pressures and blood flows in different parts of the vascular bed were described by a system of ODEs. The compliances, hydraulic and inertial resistances of aorta, large systemic and pulmonary arteries and veins, and the resistances of the systemic and pulmonary microcirculation were parameters of these equations. These parameters were set up to describe the characteristic values and the time course of the pressures, volumes, and blood flows in different parts of the CVS of healthy humans. The model also accounted for changes in the inertial and hydraulic resistance upon constrictions of the left ventricular outflow tract and was capable of describing the changes in pressures caused by aortal valve stenosis of different severity. The model was described in detail previously [17] (see also Appendix A and Supplementary Materials Table S3).

### 2.4. Numerical Simulation and the Model Validation

The finite element (FE) method formulated in small increments was implemented to solve the problem, as described in detail [16,17]. Triangular FEs with linear displacement interpolation were used. The incompressibility equations were set up and solved for every two triangles connected into a quadrilateral element.

The CVS model used here was validated thoroughly. Our cell-level model of myocardium reproduced a large set of uniaxial experiments describing tension time courses and calcium transients in isometric twitches and load-dependent relaxation in mixed isometric/isotonic modes of contraction correctly. Validation of the model was presented in detail [18]. Our CVS model, the choice of the parameters for the hemodynamics block of the model and the comparison of hemodynamic values (ventricle and atrial pressures and volumes, pressures in different vessels, blood flow through the mitral valve) was discussed and validated [17]. Not only did the model reproduce typical time-courses of hemodynamical values in healthy humans correctly, but it also was successfully used to simulate the aortic and mitral valve stenosis and insufficiency. The results of the numerical research matched the data of clinical guidelines for the classification of the valves pathologies quantitatively. The numerical methods implemented here are commonly used and have been validated for the convergence, which was checked by variation of the time-step and the size and number of the FEs [16,17]. Local strains of our simulated axisymmetric left ventricle fit clinical data [16].

### 2.5. Modeling Cell-Level Effects of Two Cardiomyopathy-Associated Mutations

We assumed that the Asp230Asn and Ile284Val Tpm mutations did not affect the kinetics, the unitary force, and the unitary myosin displacement during its interaction with actin. The kinetics of $Ca^{2+}$ release and uptake in myocardial cells was also assumed to be the same as in normal myocardium.

Only the parameters of the equations that describe $Ca^{2+}$ binding to Tn, strain-dependence of the binding, and the number of myosin filaments per cross-section area in the myocardial model were changed to simulate the effects of the mutations.

Sequeira et al. [15] studied the $Ca^{2+}$–force relationship in single permeabilized myocardial cells from the LV of a patient with HCM-associated Ile284Val Tpm mutation and healthy donors. They had found that maximal tension at saturating $Ca^{2+}$ concentration decreased by approximately 55% upon the Ile284Val Tpm mutation, while the $Ca^{2+}$ concentration required for half-maximal activation decreased by a third. Besides, the length-dependent activation [19] in the cells from the HCM patient was significantly reduced as compared to that in healthy donors [15]. The maximal sliding velocity of reconstructed thin filaments containing Ile284Val Tpm in vitro was the same as of those with wild-type (WT) Tpm [20]. To simulate these experimentally observed changes, we decreased the density of myosin filaments per cross-section area (leading to a decrease in the maximal active tension) and the parameter of length-dependency of the activation while increasing the parameter of the Tn affinity for $Ca^{2+}$ (thus increasing $Ca^{2+}$ sensitivity). The model parameters and changes introduced to simulate the effects of the Tpm mutations listed above are described in Supplementary Materials Table S4. The resulting dependencies of isometric tension and the activation level in the overlap zone $A_1$ on dimensionless $Ca^{2+}$ concentration are shown in Figure 1 (red).

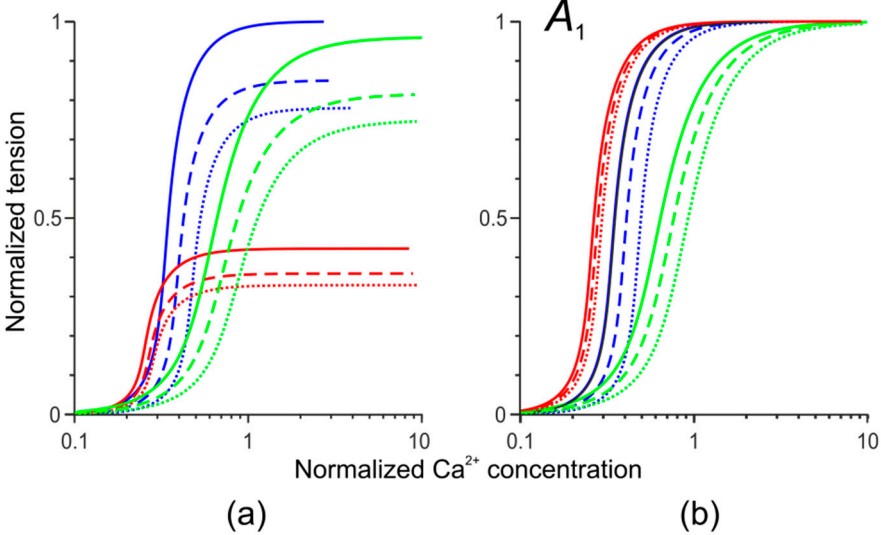

**Figure 1.** The dependence of the normalized isometric tension (**a**) and thin filament activation $A_1$ (**b**) on the dimensionless concentration of $Ca^{2+}$ ions for the simulation of normal cardiac muscle (blue) and of those with the Ile284Val (red) and Asp230Asn (green) tropomyosin (Tpm) mutations. Continuous, dashed, and dotted lines correspond to sarcomere lengths of 2.2 µm, 2.0 µm, and 1.8 µm, respectively.

We could not find in the literature detailed characteristics of changes in the $Ca^{2+}$ regulation and mechanical properties of human cardiac muscle caused by the Asp230Asn Tpm mutation. Several individuals from two families carrying the mutation have shown a mild or severe heart failure with the ejection fraction reduced down to 20% and significantly increased end-diastolic diameter of the left ventricle [13]. In vitro studies of the effects of the Asp230Asn Tpm mutation on $Ca^{2+}$ regulation of myosin ATPase in the presence of regulated thin filaments and $Ca^{2+}$ binding to thin filaments have shown a reduced $Ca^{2+}$ sensitivity and decreased $Ca^{2+}$ affinity for the thin filaments [13]. LV mechanics in vivo and in situ at the cellular and molecular levels were studied in transgenic mice carrying the Asp230Asn mutation [14]. A significantly reduced $Ca^{2+}$ sensitivity and the cooperativity of $Ca^{2+}$ regulation were found in vitro in the presence of the Asp230Asn Tpm mutation compared to WT Tpm [14,21]. To simulate the experimentally observed changes, we varied two model parameters in the regulation block of our model: the cooperativity parameter and the Tpm affinity for $Ca^{2+}$. The details are described in Supplementary Materials Table S4. The effects of these changes on the calculated

dependencies of isometric tension and the activation level on $Ca^{2+}$ in the overlap zone $A_1$ are shown in Figure 1 (green).

## 2.6. Modeling LV remodeling for HCM and DCM-Associated Tpm Mutations

The approximating geometry for the left ventricle shape was set up by the expressions from [22]. A detailed description of the approximation and the algorithm used to search for the unloaded initial configuration were described previously [16].

Sequeira et al. [15] reported that the maximal end-diastolic thickness of the septal wall of the hypertrophic left ventricle of the patient with the Ile284Val Tpm mutation was about 16 mm, while in healthy donors, it was 13 mm. The average end-diastolic thickness of the ventricle wall in a large population of healthy people was approximately 7 mm in men and 6 mm in women [23] with maximal septal wall thickness being, on average, 8.2 mm in men and 6.9 mm in women. It was also reported [24,25] that the end-systolic volume of the hypertrophic LV and the short radius of its cavity are, generally, slightly decreased.

In order to simulate the remodeling accompanying HCM according to the above cited data, the inner and outer end-systolic basal radii and the thickness of the ventricular apex were changed as described in Supplementary Materials (Table S5, HCM). These changes resulted in the increase in the end-diastolic LV wall thickness to 11 mm from 6 mm in the model of the normal LV.

To simulate ventricular dilatation caused by the Asp230Asn Tpm mutation, we changed the near-end-systolic geometry in accordance with the data reported for the transgenic mice [14]. In particular, the parameter choice was based on the relative value of the increase in the end-diastolic size and volume of the LV cavity of the mice. The changes in the model parameters made to simulate DCM caused by the mutation are specified in the Supplementary Materials (Table S5, DCM).

## 2.7. Modeling Changes in Passive Myocardial Stiffness Accompanying HCM and DCM

In the simulation of the LV remodeling described above (Section 2.6), no changes in the passive myocardial stiffness accompanying HCM or DCM were taken into account. There are no data on the passive properties of the LV myocardium associated with the Tmp mutations simulated here. However, there are clinical research data for similar HCM and DCM cases. A decrease in the titin-based stiffness in patients with the end-stage heart failure caused by nonischemic DCM was found [5]. These changes correlated with an increased expression of the long N2BA titin isoform in DCM myocardium as compared to its expression in normal myocardium, where the shorter N2B isoform was mainly expressed. Interestingly, no changes in the strain–stress diagram were found after removing the titin stress component by high ionic strength solutions. To reproduce these DCM data in our model, we increased the contour titin length [26] from 0.35 µm in normal myocardium to 0.725 µm in DCM myocardium, leaving the isotropic stress–strain relation the same as in normal myocardium.

In clinical research, the myocardial stiffness estimated by the measurement of shear wave velocity was 2–3 times higher in HCM patients than in healthy volunteers [7]. In rats with LV hypertrophy caused by aortic banding, titin stiffness increased significantly, while non-titin components showed only the slightest increase [6]. As the data are controversial, we tested the effects of an increase in each of two components of passive myocardial stiffness: a twofold increase in the isotropic extracellular stiffness or a decrease in titin contour length (0.24 µm instead of 0.35 µm in normal myocardium model).

## 3. Results

### 3.1. Simulation of Hemodynamic Changes Caused by the Asp230Asn and Ile284Val Tpm Mutations Without the LV Remodeling

To understand how the cell-level changes in mechanical properties of cardiac muscle caused by the Tpm mutations might affect the heart function, we simulated steady-state heartbeats at a rate of 60 heartbeats per min by the model with the default 'normal' LV size and shape [16,17], while the

model parameters of $Ca^{2+}$ regulation were changed as described in Section 2.5 (see also Supplementary Materials, Table S4) and shown in Figure 1. All other parameters of the model were the same as those for the normal heart model. The simulations performed with the model parameters corresponding to the normal right ventricle showed blood redistribution from the pulmonary circulation to the systemic one and a significant LV overfill. To overcome this problem in the absence of available data regarding changes in the right ventricle geometry and function caused by the mutations, we decreased the parameter of maximal isovolumetric pressure for the right ventricle to obtain the same end-diastolic LV volume as that in the normal heart model. The parameter was decreased from 85 to 70 mm Hg for the simulation of the Ile284Val Tpm mutation and to 60 mm Hg for the case of the Asp230Asn one. The duration of the right ventricle systole remained unchanged for simplicity. The hemodynamic variables obtained during the heartbeat simulations are shown in Figure 2.

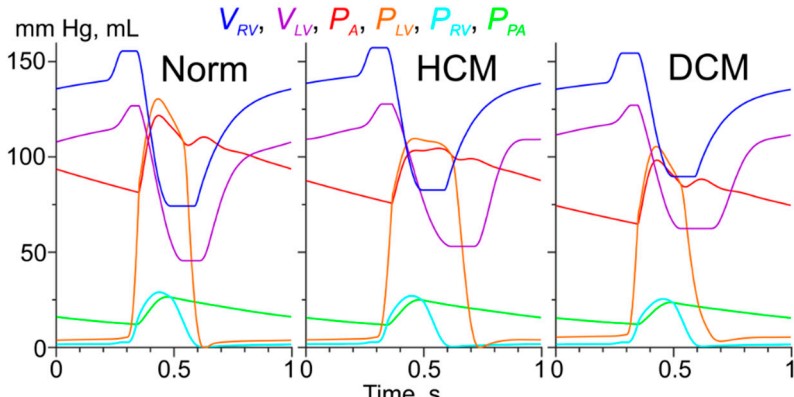

**Figure 2.** The results of the simulations of heartbeats for model of the normal (Norm) left ventricle (LV) cardiac muscle and those with the Ile284Val (hypertrophic cardiomyopathy, HCM) and Asp230Asn (dilated cardiomyopathy, DCM) Tpm mutations. $V_{RV}$, $V_{LV}$ are volumes of the right and left ventricles, respectively; $P_A$, $P_{LV}$, $P_{RV}$, and $P_{PA}$ are pressures in aorta, left and right ventricles and pulmonary artery, respectively. The color code is shown on top of the plots.

The end-diastolic and end-systolic LV volumes for the model of the normal heart were 127 mL and 45 mL, respectively, while the ejection fraction was 65%. These hemodynamic characteristics as well as the systolic (121 mm Hg) and diastolic (81 mm Hg) aortic pressures were close to those reported for healthy humans. The simulation of heartbeats in the presence of the Ile284Val Tpm mutation associated with HCM showed a mild reduction in the heart performance: the stroke volume and the rejection fraction reduced to 75 mL and 59%, respectively, while the systolic and diastolic aortic pressure decreased to 105 mm Hg and 75 mm Hg, respectively (Figure 2, HCM). The simulations of the effect of the Ile284Val Tpm mutation also showed a prolongation of LV systole from 189 ms in the normal heart model to 253 ms. Simulation of the effects of the Asp230Asn Tpm mutation resulted in more severe hemodynamic changes. The stroke volume and the LV ejection fraction reduced to 65 mL and 51%, respectively. The aortic blood pressure was also decreased compared to that in the simulations with the normal LV and was equal to 98/65 mm Hg (Figure 2, DCM).

## 3.2. Changes in $Ca^{2+}$ Transients Caused by the Tpm Mutations

The changes in the cell-level model parameters caused by the Tpm mutations affected the time course of the intracellular variables describing myocardial activation, as shown in Figure 3.

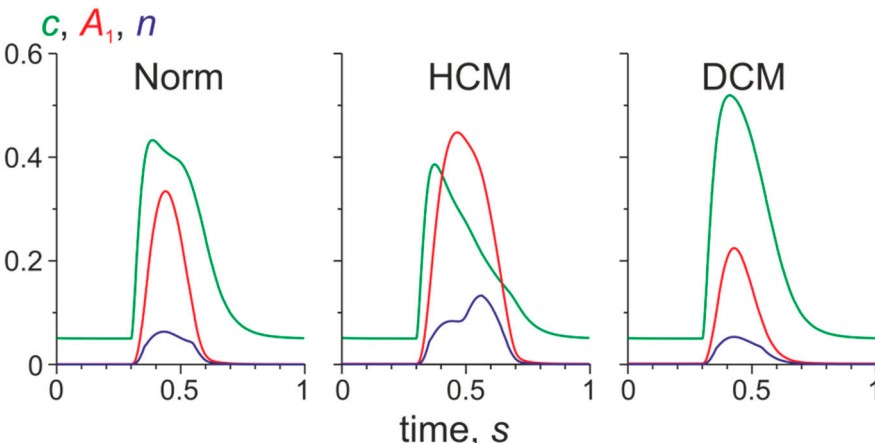

**Figure 3.** The time course of the normalized $Ca^{2+}$ concentration in the cytosol ($c$), the level of activation of the Tpm–Tn (troponin) system in the overlap zone of sarcomeres ($A_1$), and the fraction of myosin heads bound to actin ($n$) in a mid-wall finite element located near the LV equator. The results obtained in the normal LV model (Norm) and those with normal LV geometry and cell-level parameters characteristic for the HCM and for DCM Tpm mutations are shown.

In the HCM model, the peak systolic values of $A_1$ and $n$ increased compared to their normal values, while the $Ca^{2+}$ peak decreased (Figure 3, HCM). These changes are caused by the shift of the force-$Ca^{2+}$ toward lower $Ca^{2+}$ concentration (Figure 1) and by a reduction of the free intracellular $Ca^{2+}$ concentration due to its binding to Tn. In contrast, in the DCM model, the peaks of $A_1$ and $n$ were reduced, while the peak of the free $Ca^{2+}$ concentration was enhanced (Figure 3, DCM).

### 3.3. Simulation of Hemodynamic Changes Caused by the Tpm Mutations and the LV Remodeling

The end-diastolic and end systolic shapes of the normal model LV and those with DCM and HCM are shown in Figure 4.

The model of dilated LV with the Asp230Asn Tpm mutation showed higher end-diastolic and end-systolic volumes than those of the normal LV (Figure 4). The fiber strain in the DCM LV model was noticeably smaller than in the normal LV model. In contrast to DCM that caused more uniform distribution of sarcomere length than in the normal LV model, the model of HCM LV was characterized by a higher transmural difference in sarcomere length. This resulted in very short sarcomeres at the end of systole in the subepicardium.

The twist of the LV apex with respect to the base for the model of normal LV between systole and diastole was 16.3°. It decreased to 6.4° for the DCM model and increased to 37.9° for the HCM model. The higher twist may be responsible for the sarcomere length heterogeneity observed in the HCM simulation (Figure 4). We have also measured the global longitudinal strains (GLS) of the simulated LVs with the standard procedure used in 2D echocardiography. In simulations, GLS was decreased moderately in HCM LV (−15.1%) and drastically in DCM LV (−11.1%) compared to the value of approximately 18.7% for normal LV. The animations showing the changes in the LV shape during a heartbeat are given in the Supplementary Materials (Video S1).

The results of the simulation of the effects of the Tpm mutation and the LV remodeling (dilatation for the Asp230Asn mutation and hypertrophy for the Ile284Val one) on systemic and pulmonary hemodynamics are shown in Figure 5.

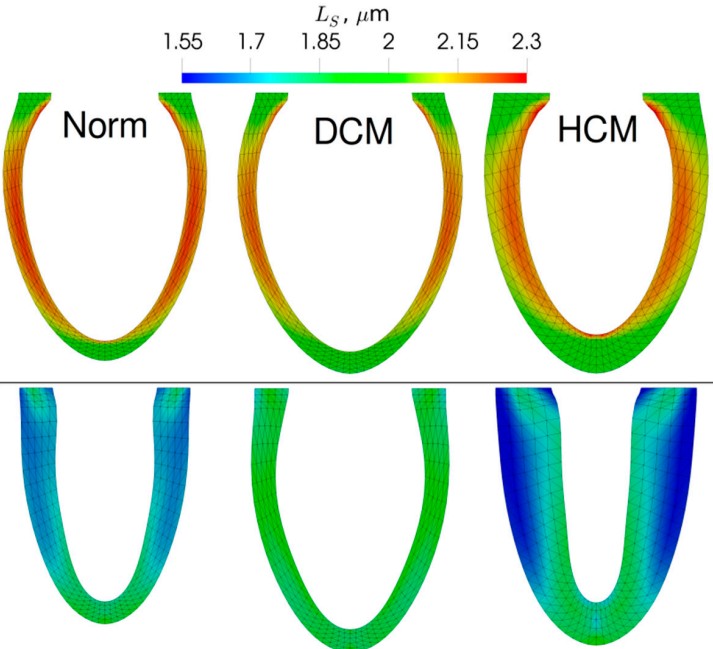

**Figure 4.** The end-diastolic (top) and end-systolic (bottom) LV geometry for the models of the normal (Norm), dilated (DCM), and hypertrophic (HCM) LV obtained during the simulations. The color code shows sarcomere length.

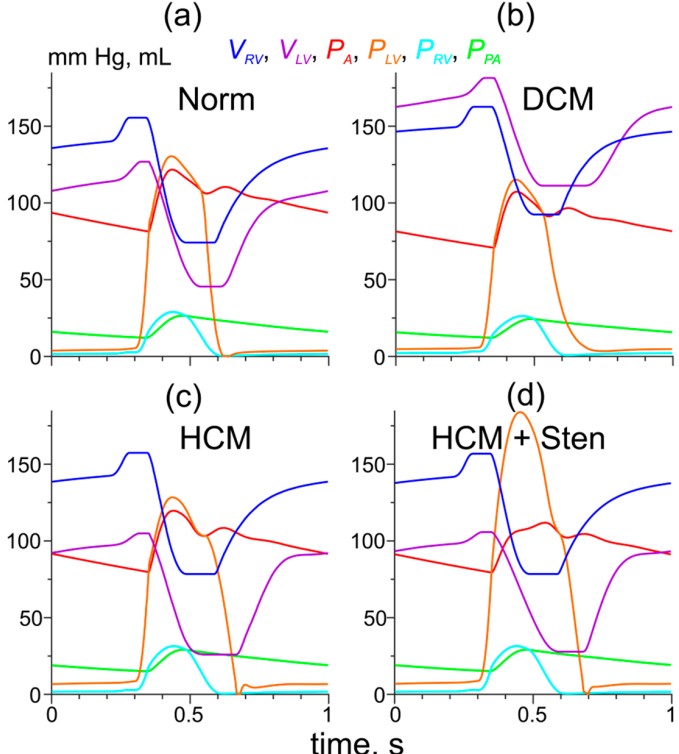

**Figure 5.** The results of simulations of hemodynamic variables during a heartbeat for the model of normal LV myocardium and normal LV geometry (**a**) and those with the DCM- and HCM-associated Tpm mutations and LV remodeling (**b,c,d**). (**b**) The cell level effects of the Asp230Asn TPM mutation were combined with LV dilatation as described in Methods; (**c**) the cell-level effects of the Ile284Val Tpm mutation were combined with LV hypertrophy as described in Methods; (**d**) the same as (**c**) plus the stenosis of the LV outflow tract. The color codes for the pressures and volumes are shown.

An increase in the volume of the LV cavity (dilatation) in the model with the Asp230Asn Tpm mutation led to partial compensation of a decrease in the LV performance caused by the changes in $Ca^{2+}$ sensitivity (Figure 5b). The stroke volume and the aortic blood pressure reduced, although remaining close to those in the normal heart model (70 mL, 107/71 mm Hg versus 82 mL and 121/81 mm Hg, respectively) while the ejection fraction was reduced significantly to 39% (compared to 65%).

The LV hypertrophy was accompanied by an increase in passive LV stiffness: at a 'normal' end-diastolic LV pressure (EDLVP) of 5.3 mm Hg, the LV volume was only 92 mL. Our simulation of the remodeled LV was performed with the 'normal' initial values of the hemodynamic variables (excluding the LV volume) and the 'healthy' right ventricle (default values of $E_{3RV}$, Table S3). Such conditions resulted in an increased LV preload of 8.6 mm Hg and the end-diastolic volume of 105 mL. Simulation of the effect of LV hypertrophy for the HCM-associated Ile284Val Tpm mutation also resulted in a more complete recovery of the hemodynamic parameters (Figure 5c). The aortic pressure of 120/80 mm Hg and the stroke volume of 79 mL were close to those calculated for the normal heart model (121/81 mm Hg, 82 mL), while the ejection fraction increased to 75%. When the stenosis of the LV output tract (maximal orifice area 1 $cm^2$ at aortic area of 3 $cm^2$) was taken into account in the model of HCM caused by the Tpm mutation, the peak LV systolic pressure increased up to 184 mm Hg, while the aortic pressure (112/79 mm Hg) and the stroke volume (78 mL) reduced slightly compared to the HCM model without the stenosis (Figure 5d). The peak pressure gradient (72 mm Hg) was similar to that in a patient with this mutation [15].

As one would expect, a decrease in titin stiffness in the LV myocardium at DCM led to a slight increase in the end-diastolic LV volume at fixed values of end-diastolic pressure and to an increase in the stroke volume and arterial blood pressure. An increase in any component of the passive myocardial stiffness in the LV myocardium at HCM led to a further impairment of LV diastolic function and decreased LV performance (Table 1).

**Table 1.** The effects of passive myocardial stiffness on the model LV performance at HCM and DCM.

| LV Hemodynamic Characteristics | HCM + Normal Passive Stiffness | HCM + Stiff Titin Component | HCM + Stiff Isotropic Component | DCM + Normal Passive Stiffness | DCM + Soft Titin Component |
|---|---|---|---|---|---|
| **Low Preload** | | | | | |
| End-diastolic pressure, mm Hg | | 6.4 | | 4.3 | |
| Peak pressure, mm Hg | 115.2 | 112.6 | 92.5 | 105.4 | 108.5 |
| End-diastolic volume, mL | 92.2 | 88.3 | 73.2 | 157.5 | 165.1 |
| Stroke volume, mL | 67.7 | 64.1 | 48.3 | 64 | 66.1 |
| Ejection fraction, % | 73 | 73 | 66 | 41 | 40 |
| **Average Preload** | | | | | |
| End-diastolic pressure, mm Hg | | 8.7 | | 5.1 | |
| Peak pressure, mm Hg | 128.4 | 123.1 | 99.3 | 110.7 | 113.2 |
| End-diastolic volume, mL | 104.9 | 99.2 | 80 | 169.8 | 176.6 |
| Stroke volume, mL | 79 | 74 | 54.8 | 67.3 | 68.9 |
| Ejection fraction, % | 75 | 75 | 69 | 40 | 40 |

**Table 1.** *Cont.*

| LV Hemodynamic Characteristics | HCM + Normal Passive Stiffness | HCM + Stiff Titin Component | HCM + Stiff Isotropic Component | DCM + Normal Passive Stiffness | DCM + Soft Titin Component |
|---|---|---|---|---|---|
| **High Preload** | | | | | |
| End-diastolic pressure, mm Hg | | 11.5 | | | 5.8 |
| Peak pressure, mm Hg | 141.1 | 133 | 108.6 | 115.2 | 118.3 |
| End-diastolic volume, mL | 117.9 | 109.5 | 88.8 | 181.5 | 191 |
| Stroke volume, mL | 89.9 | 83.1 | 63 | 70.2 | 72.1 |
| Ejection fraction, % | 76 | 76 | 71 | 39 | 38 |

### 3.4. Simulation of the Effects of the Tpm Mutations and the LV Remodeling on the Pressure-Volume Loops

The LV pressure–volume loops (PV-loops) obtained at different preloads are often used to estimate the systolic and diastolic functions of the heart chambers. In particular, the slope of the line plotted through the end-systolic point of the loops, the so-called end-systolic pressure–volume relationship (ESPVR) is believed to characterize the LV contractility [27]. Figure 6 shows the PV-loops obtained in the simulations of normal and cardiomyopathic LVs. To probe the LV performance at various preloads, initial blood pressures in systemic and pulmonary veins were varied.

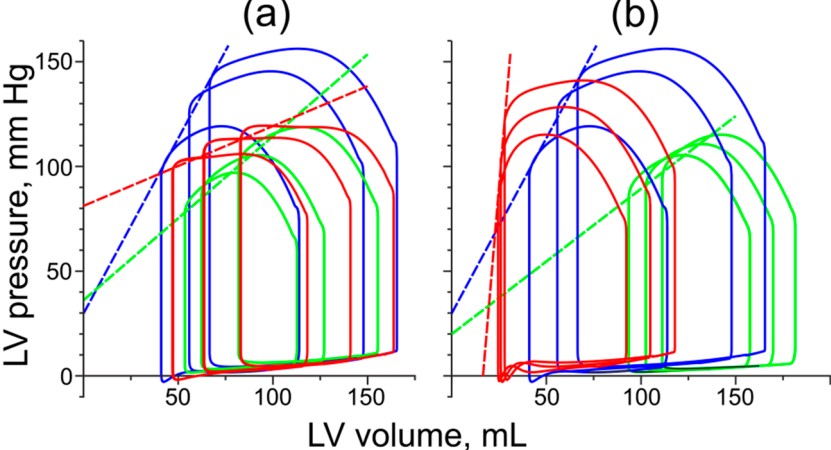

**Figure 6.** The LV pressure–volume loops obtained from the simulation of the LV with normal cardiac muscle and normal geometry (blue) and the simulations of DCM (green) and HCM (red). Different loops were obtained at different preloads. (**a**) PV loops obtained with normal LV geometry; (**b**) PV loops obtained for the remodeled LVs with DCM and HCM at default passive myocardial stiffness. The dashed straight lines are the ESPVR lines plotted for each simulation case.

Both HCM and DCM resulted in a decrease in the slope of ESPVR if the LV remodeling was not considered, and only changes in myocardial properties caused by the Tpm mutations were taken into account (Figure 6a. This agrees with the idea that the slope characterizes myocardial contractility [28]: the DCM-associated mutation decreases the $Ca^{2+}$ sensitivity of active tension, while the HCM-associated mutation decreases the maximal active tension and length-dependent activation (Figure 1).

LV dilatation led to a further decrease in the ESPVR slope compared to the effect of the Asp230Asn Tpm mutation alone (Figure 6b, green). The slope of ESPVR in the DCM simulations decreased by a factor of approximately 2.5 compared to the normal LV model. On the contrary, the LV hypertrophy (Figure 6b, red) dramatically increased the ESPVR slope compared to that in the absence of hypertrophic

remodeling and even compared to the slope in the simulations of the normal heartbeat, despite impaired myocardial properties.

## 4. Discussion

### 4.1. Cell-Level Changes in $Ca^{2+}$ Regulation and Cardiac Function

When the molecular-level effects of the Asp230Asn Tpm mutation [13,14] were introduced into our myocardial model, the amplitude of $Ca^{2+}$ transient increased as was observed in transgenic mice [14], while the activation level ($A_1$) and the fractions of actin-bound myosin heads ($n$) decreased compared to the model of normal heart (Figure 3, DCM). When changes in $Ca^{2+}$ regulation found in myocardial cells with the HCM-associated Ile284Val Tpm mutation [15] were simulated, the amplitude of $Ca^{2+}$ transient decreased, while the fraction of actin bound myosin heads increased (Figure 3, HCM). These effects result from a combination of several factors: change in $Ca^{2+}$ sensitivity of the thin filaments caused by the mutations (Figure 1) and $Ca^{2+}$ binding to Tn affecting the free $Ca^{2+}$ concentration. Despite the difference, changes in the cell-level model parameters corresponding to those caused by both the HCM and DCM Tpm mutations decreased arterial blood pressure, stroke volume, and ejection fraction compared to the model of normal LV (Figure 2).

### 4.2. Effects of LV Remodeling

To simulate the effect of the remodeling accompanying DCM or HCM, we changed the LV geometry in the model according to available data for the Asp230Asn and Ile284Val Tpm mutations, respectively.

The LV dilatation (Figure 4, DCM) partially, although not completely, compensated for the loss of the stroke volume caused by the Asp230Asn Tpm mutation (Figure 5b). In this case, a partial restoration of the heart function was accompanied by a further reduction in the ejection fraction to a value close to those found in members of two families with the mutation [13]. Compensation of the stroke volume at a decreased ejection fraction was also observed in transgenic mice [14]. We suppose that the compensation occurred due to the increase in the end-diastolic volume of the LV, which provided the normal stroke volume despite the Starling's law of the heart estimated by the PV loops being violated (Figure 6b green).

A decrease in passive myocardial stiffness due to an increase in the contour length of titin in the model had only a slight effect on the heart performance, enhancing the compensatory effect of LV dilation a little (Table 1). The results of the simulation show that the changes in cardiac titin observed in myocardial samples from DCM patients [5] might be a part of the LV remodeling that helps preserve its function.

When the LV hypertrophy that accompanied the Ile284Val Tpm mutation was included in the LV model, it resulted in nearly full compensation of the stroke volume and the aortic blood pressure at preserved initial hemodynamic conditions, which provided an increased preload of the LV. At the 'normal' values of the preload, no compensation was observed. When stenosis of the outflow tract, which accompanied the LV hypertrophy in a patient with the Ile284Val Tpm mutation [15], was accounted for, the stroke volume and the ejection fraction were close to their values for the model of normal LV at elevated preload (Figure 5c,d). The results of our simulations suggest that the wall thickening is able to compensate for the reduction of active force and the impaired length-dependent $Ca^{2+}$ activation caused by the Ile284Val Tpm mutation. However, an increase in the isotropic or anisotropic (titin) component of passive myocardial stiffness along with the wall thickening led to an even more severe impairment of LV diastolic and systolic function (Table 1). The hypertrophy also led to heterogeneity in sarcomere length distribution across the LV wall (Figure 4) and to an increase in the LV twist.

Significant values of the decrease in the LV twist and absolute values of GLS in our simulation of the DCM LV were similar to those observed in DCM patients [29–31]. The decrease in the LV GLS obtained here was also close to that in DCM patients [30–32]. An increase in the LV twist compared to

control was observed in patients with HCM. The observed increase varied, being either significantly lower than that calculated here [33,34] or close to it [30]. The decrease in the calculated GLS was also similar to that in HCM patients [30,34–36]. An increase in the isotropic (extracellular) or anisotropic (titin) component of the passive myocardial stiffness in the HCM LV model worsened the LV function by impairing its diastolic function (Table 1). The increase in passive stiffness together with the thickening of the LV wall itself are the factors impairing LV diastolic function.

### 4.3. PV Loops Depend on Not Only on Myocardial Contractility but Also LV Geometry

Our simulation of PV loops for the normal and cardiomyopathic LVs resulted in the following conclusions and suggestions.

Firstly, the ESPVR slope does characterize the contractile properties of cardiac muscle at fixed LV geometry, as was initially suggested by Suga and Sagawa [27]. The simulated ESPVRs had lower slopes for the cases of both HCM-and DCM-associated mutations in not remodeled LV compared to those for the normal LV model (Figure 6a). The decrease in the ESPVR slope was caused by the impaired properties of the muscle: the decreased $Ca^{2+}$ sensitivity for the DCM mutation and the decreased maximal force with a simultaneous reduction in the length-dependence of $Ca^{2+}$ activation for the HCM mutation.

Secondly, in the simulations of the dilated LVs with the Asp230Asn Tpm mutation, the ESPVR slope was even lower than in the LV with the same mutation and normal geometry. The decrease in the ESPVR slope by a factor of 2.5 in the simulations of the DCM LVs compared to the model of normal LV is similar to that observed [14] for WT and transgenic mice. A possible explanation for the further reduction in ESPVR is as follows. An enlargement of the LV cavity at a normal wall thickness leads to a decrease in the end-systolic myocardial stress at the same end-systolic blood pressure.

Thirdly, the LV hypertrophy that accompanies the Ile284Val Tpm mutation (in contrast to the effects of the LV dilatation) led to an even steeper ESPVR slope than that obtained for the normal LV model (Figure 6b. We were not able to find any PV-loop data for this particular mutation, but an increase in the ESPVR slope for the patients with heart failure, concentric HCM, and non-impaired ejection fraction was reported [37,38]. We suppose that the increase in the ESPVR slope for the HCM LV, compared to that with the Ile284Val Tpm mutation and normal geometry, was caused by an increase in the thickness of the LV wall and, possibly, by the enhanced $Ca^{2+}$ sensitivity of cardiac muscle. From our results and published clinical data, we can suggest that the ESPVR slope is strongly affected by changes both in the contractile properties of the myocardium and the LV geometry and cannot be used as a characteristic of myocardial contractility for hypertrophic LVs.

### 4.4. Relation to Previous Works

A number of models were suggested to simulate heart work in health and disease. A simple 0D lumped parameter model [39] was used to simulate the heartbeats of normal and DCM hearts. In this model, the myocardial dysfunction was described by a decrease in the end-systolic LV elastance, which is a value that is difficult to relate to the specific changes in cardiac muscle characteristics underlying DCM. A multiscale model of the CVS containing an accurate cell-level description of myocardial electromechanics and simple thin-wall spherical approximation of the ventricles [40] was applied to study the ventricular contractility at different preload and calcium kinetics in myocytes. The model was further validated [41], being able to reproduce some experimental data including the effects of an anesthetic on hemodynamics through its influence on the $Ca^{2+}$ sensitivity of myofilaments. A more sophisticated 3D electromechanical model of a DCM heart was used to estimate the efficiency of an LV-assist device [42]. Although these authors used more realistic anatomy of the ventricles than the one in our model and included electrophysiological processes, the description of the active tension was too simplified to reproduce the changes in active tension and its regulation caused by the mutations. Several models were suggested to describe heart remodeling (see reviews [43,44]). Some models of this kind describe electromechanics of failing heart including concentric and eccentric hypertrophy.

For example, model [45] described electrophysiology, ventricle anatomy, and passive myocardial mechanics in some detail, while the specification of active tension did not allow one to simulate the effects of the Tpm mutations. A 3D simulation of the diastolic function of DCM heart that considered its detailed remodeling including the changes in a number of sarcomeres in cardiomyocytes was presented [46]. Concentric hypertrophy was simulated by a detailed 3D electromechanical model [47], where cardiac muscle electrophysiology was described by a bidomain model and its mechanics were specified by a detailed cardiac cell model [48]. The authors examined the effects of heart remodeling (concentric hypertrophy) caused by aortic stenosis on heart performance and did not investigate any effects of mutations of sarcomere proteins.

### 4.5. Limitations

Our simulation was based on a simplified axisymmetric LV model. The 3D models [49] are too expensive computationally for a thorough investigation of the effects of the Tpm mutations. These effects are caused by rather uniform cell-level changes in cardiomyocytes, so that a 2D approximation appears to be sufficient for their simulation. Besides, no 3D data are available for the LV geometry of patients with the Tm mutations considered here. For these reasons, the use of a model with reduced dimensionally and detailed description of the cell-level mechanics and Ca$^{2+}$ regulation seems to be a reasonable simplification.

In our simulations, we did not take into account any alteration of the distribution of the orientation of cardiac fibers in the LV wall as a long-time effect of HCM and DCM accompanying the remodeling of the LV geometry [50,51]. This can be the reason for the discrepancy between our estimation of the LV twist upon HCM and some clinical data. The absence of the right ventricle in our finite element model may be another reason for the discrepancy in the LV twist calculation. It should also be noticed that the LV remodeling was set in a straightforward manner without the consideration of any assumptions on the kinetics of growth and/or structural rearrangement [43,44,46]. We also were not able to reproduce possible alterations in myocardium electrical excitability and conductivity including the remodeling of the gap junctions accompanying the cardiomyopathies [52], because our model did not contain a description of myocardium electrophysiology.

**Supplementary Materials:** The following are available online at http://www.mdpi.com/2227-7390/8/7/1169/s1, Table S1: Parameters of the myocardium cell model, Table S2: LV shape approximation parameters, Table S3: Parameters of the hemodynamic model, Table S4: Values of the parameters for the regulation block of the myocardium cell-model for the simulation of HCM- and DCM-associated mutations, Table S5: Geometrical parameters of the normal LV and the LVs with cardiomyopathy. Video S1: Animation of the changes in the shape and sarcomere lengths during the simulation of a heartbeat. Left to right: the normal, HCM, and DCM LV models. The animation starts with the atrial systole. The color code is for sarcomere length in μm.

**Author Contributions:** Conceptualization, F.S. and A.T.; software, F.S. and A.K.; validation, F.S.; investigation, numerical simulations and the results analysis, F.S., A.K. and A.O.; visualization—A.K. and A.O., writing—original draft preparation, F.S.; writing—review and editing, A.K., A.O. and A.T.; supervision, A.T. All authors have read and agreed to the published version of the manuscript.

**Funding:** This research was funded by the Russian Foundation for Basic Research, grant numbers 18-31-00065 (for F.S.) and 17-00-00066 (a part of a complex grant 17-00-0071, for A.T.)**.**

**Conflicts of Interest:** The authors declare no conflict of interest. The funders had no role in the design of the study; in the collection, analyses, or interpretation of data; in the writing of the manuscript, or in the decision to publish the results.

## Appendix A

### A.1. Cell Model of Myocardium [18]

Cauchy stress tensor **T** was specified by the following equation:

$$\mathbf{T} = \left( \frac{\partial \Phi(I_1, I_2)}{\partial I_1} + I_1 \frac{\partial \Phi(I_1, I_2)}{\partial I_2} \right) \mathbf{F} - \frac{\partial \Phi(I_1, I_2)}{\partial I_2} \mathbf{F}^2 - p\mathbf{E} + \mathbf{B}(T_{tit} + T_A) \qquad (A1)$$

**F** is the Finger deformation tensor, $I_1$, $I_2$ are its first and second invariants;

**E** is the unit tensor, $p$ is the Lagrange factor caused by incompressibility;

**B** is the anisotropy tensor equal to the tensor square of the unit vector $\vec{c}_f$ aligned with the direction of muscle fibers in deformed muscle $\mathbf{B} = \vec{c}_f \otimes \vec{c}_f$.

The isotropic elastic strain energy $\Phi$ was taken in the form similar to that used by Guccione et al. [53]:

$$\Phi = a_0 exp\big(a_1\big((I_1 - 3)^2 - 2(I_2 - 2I_1 + 3)\big)\big), \tag{A2}$$

where $a_0$, $a_1$ are constant parameters. The difference from the work [53] is the absence of anisotropic part, which, in our model, was included only in the last term of the Equation (A1).

$T_{tit}$ is the anisotropic passive tension of the intra-sarcomere cytoskeleton mainly caused by titin filaments; $T_A$ is the active tension produced by the actin–myosin interaction. Titin tension was specified by the equation based on the worm-like chain model [26].

$$T_{tit} = \begin{cases} N_M \frac{6k_B T}{L_p}\left(\frac{1}{4}\left(1 - \frac{L_s - L_{s0}}{L_c}\right)^{-2} + \frac{L_s - L_{s0}}{L_c} - \frac{1}{4}\right), L_s \geq L_{s0}, \\ N_M \frac{9k_B T}{L_p L_c}(L_s - L_{s0}), L_s < L_{s0}. \end{cases} \tag{A3}$$

$L_s$ and $L_{s0}$ are the deformed and reference sarcomere lengths, $N_M$ is the number of the myosin filaments per unit cross-section area of muscle in its initial reference state; $L_c$ is the total, or 'contour', length of a titin molecule, $L_p$ is so-called persistence length; $k_B$ and $T$ are the Boltzmann constant and absolute temperature. The deformed sarcomere length is described by the equation $L_s = L_{s0}\sqrt{\vec{c}_{f_0}\mathbf{G}\vec{c}_{f_0}}$, where **G** is the right Cauchy–Green deformation tensor, $\vec{c}_{f_0}$ is the unit vector aligned with fibers in unstrained muscle.

Cross-bridge kinetics was based on the Lymn–Taylor cycle, in which a specific part of a myosin molecule (myosin head) can be in a free state or attached to actin filament in two different ways: a weakly bound and force generating strongly bound state. Thus, the active tension was specified as $T_A = E_{cb}N_M N_{cb}W(L_s)n(\delta + \theta h)$. Here, $N_{cb}$ is the total number of myosin heads per one-half of a myosin filament; $W(L_s)$ is the length of the overlap zone of the thick and thin filaments in a half-sarcomere normalized for its maximal value; $E_{cb}$ is the constant cross-bridge stiffness, $n$ is the probabilities of a myosin head being attached to the actin filament, $\theta$ is the fraction of strongly bound cross-bridges among $n$, and $h$ is a cross-bridge distortion during transition from the weakly bound state to the strongly bound one. The kinetic equations for $n$ and $\theta$ are as follows

$$\frac{\partial n}{\partial t} = f_+(\delta)(A_1 - n) - f_-(\delta)n, \tag{A4}$$

$$\frac{\partial n\theta}{\partial t} = H_+(\delta)n(1 - \theta) - H_-(\delta)n\theta \Rightarrow (H_+, H_- \gg f_+, f_-) \Rightarrow \theta = \frac{H_+(\delta)}{H_+(\delta) + H_-(\delta)}. \tag{A5}$$

Here, $f_+, f_-, H_+, H_-$ are kinetic rates that depend on ensemble-averaged cross-bridge distortion $\delta$. The equation for the normalized cross-bridge distortion $\delta' = \delta/h$ was

$$\frac{\partial \delta'}{\partial t} = \frac{1}{2h}\frac{\partial L_s}{\partial t} - \frac{(A_1 - n)}{n}f_+(\delta')\delta', \tag{A6}$$

and the kinetic rates were set as

$$f_+(\delta') = f_+^0 \begin{cases} 1, \delta' \leq 0, \\ \frac{\delta_0'^2}{(\delta_0 - \delta')^2}, \delta' > 0, \end{cases}, \quad f_-(\delta') = f_+^0 \begin{cases} b_{cb} + c_{cb}\delta'^2, \delta' \leq 0 \\ b_{cb} + \frac{\delta'}{\delta_0 - \delta'}, \delta' > 0, \end{cases} \frac{H_+(\delta')}{H_-(\delta')} = e^{-\Delta\delta'}. \tag{A7}$$

In Equation (4), $A_1$ is the probability that a binding site of actin in the overlap region of the actin and myosin filaments is in the open state for the myosin head. The similar probability for the region outside the overlap zone was denoted by $A_2$. The kinetics of these variables depended on $c$ (Ca$^{2+}$ concentration in cytoplasm normalized by 'normal' half-activation concentration $c^0{}_{50}$), cooperativity parameter $m$ (the Hill coefficient), sarcomere length via parameter $k_s$, and the number of strongly bound cross-bridges via parameter $k_n$. $W_a$ is the length of the overlap zone normalized by the actin length per half sarcomere.

$$\frac{\partial A_1}{\partial t} = \begin{cases} \alpha_+\left(c(1-A_1)^{\frac{1}{m}} - \frac{c_{50}A_1^{\frac{1}{m}}}{(1+k_s(L_s-L_{s0})/L_{s0})(1+k_n n\theta)}\right), & \frac{\partial W_a(L_s)}{\partial t} \le 0, \\ \alpha_+\left(c(1-A_1)^{\frac{1}{m}} - \frac{c_{50}A_1^{\frac{1}{m}}}{(1+k_s(L_s-L_{s0})/L_{s0})(1+k_n n\theta)}\right) + \frac{\partial W_a(L_s)}{\partial t}\frac{(A_2-A_1)}{W_a(L_s)}, & \frac{\partial W_a(L_s)}{\partial t} > 0, \end{cases} \tag{A8}$$

$$\frac{\partial A_2}{\partial t} = \begin{cases} \alpha_+\left(c(1-A_2)^{\frac{1}{m}} - \frac{c_{50}A_2^{\frac{1}{m}}}{(1+k_s(L_s-L_{s0})/L_{s0})}\right) + \frac{\partial W_a(L_s)}{\partial t}\frac{(A_2-A_1)}{(1-W_a(L_s))}, & \frac{\partial W_a(L_s)}{\partial t} \le 0, \\ \alpha_+\left(c(1-A_2)^{\frac{1}{m}} - \frac{c_{50}A_2^{\frac{1}{m}}}{(1+k_s(L_s-L_{s0})/L_{s0})}\right) + \frac{\partial W_a(L_s)}{\partial t}, & \frac{\partial W_a(L_s)}{\partial t} > 0, \end{cases} \tag{A9}$$

$$\left(1 + \frac{k_{BC}B_{Ca}}{(c \cdot c^0_{50}+k_{BC})^2}\right)\frac{\partial c}{\partial t} = I_{Ca}(t) - \frac{(Y_{1Ca}/c^0_{50})(c^2-c_0^2)}{c^2 \cdot c^0_{50}+Y^2_{2Ca}} - \left(C_{Tn}/c^0_{50}\right)\frac{\partial(A_1W_a(L_s)+A_2(1-W_a(L_s)))}{\partial t}. \tag{A10}$$

$\alpha_+$ is a characteristic rate constant of the Ca-troponin; $I_{Ca}(t) = I^0_{Ca}(exp(-k_{1Ca}t) - exp(-k_{2Ca}t))$ is a given inflow of Ca$^{2+}$ ions to the cell normalized by c$^0{}_{50}$. The parameters of this block of the model and their values are presented in Table S1.

### A.2. Left Ventricle Approximation

LV approximation was set by the following expressions for Cartesian coordinates $(r, z)$ through the curvilinear coordinates $(\gamma, \psi)$ with parameters $\varepsilon$, $r_{in}$, $r_{out}$, $h_{in}$, and $h_{out}$ [16,22].

$$\begin{aligned} r &= (r_{in} + \gamma(r_{out} - r_{in}))(\varepsilon cos\psi + (1-\varepsilon)(1-sin\psi)), \\ z &= (h_{in} + \gamma(h_{out} - h_{in}))(1-sin\psi) + (1-\gamma)(h_{out} - h_{in}). \end{aligned} \tag{A11}$$

### A.3. Hemodynamics Model [17]

Passive pressures of the right ventricle and the atria were described as follows.

$$\begin{aligned} P_{LA_{Pas}} &= E_{1LA} \cdot \left(e^{E_{2LA}V_{LA}(t)} - e^{E_{2LA}V_{0LA}} + \mu_{a1}\frac{\partial V_{LA}(t)}{\partial t}\right), \\ P_{RA\_Pas} &= E_{1RA} \cdot \left(e^{E_{2RA}V_{RA}(t)} - e^{E_{2RA}V_{0RA}} + \mu_{a1}\frac{\partial V_{RA}(t)}{\partial t}\right), \\ P_{RV\_Pas} &= E_{1RV} \cdot \left(e^{E_{2RV}V_{RV}(t)} - e^{E_{2RV}V_{0RV}}\right). \end{aligned} \tag{A12}$$

Here, $P_{**\_Pas}$ are passive elastic parts of chamber pressures, $V_{**}$ are their volumes, and $V_{0**}$, $E_1$, $E_2$, and $\mu_{a1}$ are constant parameters. Subindices *LA*, *RA*, and *RV* stand for the left atrium, right atrium, and right ventricle, respectively. Active pressures of the right ventricle and the atria were found from ordinary differential equations. Due to the introduction of time delay with relaxation time $\tau$ and the analogue of force–velocity equation, those described the pressure time-course more accurately than the pressure–volume dependencies with time-dependent stiffness coefficients commonly used in other lumped parameter models.

$$\begin{aligned} \tau\frac{\partial P_{LA\_Act}(t)}{\partial t} + P_{LA\_Act}(t) &= F_{LA\_Act}(t) \cdot \left(\mu_{a2}\frac{\partial V_{LA}(t)}{\partial t} + E_{3LA}\right), \\ \tau\frac{\partial P_{RA\_Act}(t)}{\partial t} + P_{RA\_Act}(t) &= F_{RA\_Act}(t) \cdot \left(\mu_{a2}\frac{\partial V_{RA}(t)}{\partial t} + E_{3RA}\right), \\ \tau\frac{\partial P_{RV\_Act}(t)}{\partial t} + P_{RV\_Act}(t) &= F_{RV\_Act}(t) \cdot \left(\mu_v\frac{\partial V_{RV}(t)}{\partial t} + E_{3RV}\right). \end{aligned} \tag{A13}$$

Here, $P_{**\_Act}$ are active parts of chamber pressures, and $E3$, $\mu_v$, and $\mu_{a2}$ are constant parameters. $F_{**\_Act}$ depended on activation time-functions $e(t)$ and on the volumes $V$ providing the Starling's law for the atria and the right ventricle; $k_V$, $V_{Min}$, and $V_{Max}$ are the parameters for the pressure–volume relation.

$$
\begin{aligned}
F_{LA\_Act} &= e_a(t) \cdot \left( k_V + (1 - k_V) \cdot \left( \frac{V_{LA} - V_{LA\_Min}}{V_{LA\_Max} - V_{LA\_Min}} \right) \right), \\
F_{RA\_Act} &= e_a(t) \cdot \left( k_V + (1 - k_V) \cdot \left( \frac{V_{RA} - V_{RA\_Min}}{V_{RA\_Max} - V_{RA\_Min}} \right) \right), \\
F_{RV\_Act} &= e_v(t) \cdot \left( k_V + (1 - k_V) \cdot \left( \frac{V_{RV} - V_{RV\_Min}}{V_{RV\_Max} - V_{RV\_Min}} \right) \right).
\end{aligned}
\tag{A14}
$$

Activation functions were set as follows

$$
\begin{aligned}
e_a &= \begin{cases} 0.5\left(1 - \cos\left(2\frac{(t - t_a)\pi}{T_a}\right)\right), & t_a \le t < t_a + T_a \\ 0, \ otherwise, \end{cases} \\
e_v &= \begin{cases} 0.5\left(1 - \cos\left(2\frac{(t - t_v)\pi}{T_v}\right)\right), & t_v \le t < t_v + T_v \\ 0, \ otherwise, \end{cases}
\end{aligned}
\tag{A15}
$$

where $t_a$ and $t_v$ are the times of contraction initiations for the atria and ventricles, respectively; and $T_a$, $T_v$ are the systole durations.

The full system of ordinary differential equations for other hemodynamic variables is

$$
\begin{aligned}
\frac{dV_{RV}}{dt} &= Q_{iRV} - Q_{oRV}, \\
\frac{dV_{RA}}{dt} &= Q_{iRA} - Q_{iRV}, \\
\frac{dV_{LV}}{dt} &= Q_{iLV} - Q_{oLV} + C_{LV}\frac{dP_{LV}}{dt}, \\
\frac{dV_{LA}}{dt} = Q_{iLA} - Q_{iLV}, L_{iLV}&\frac{dQ_{iLV}}{dt} + R_{iLV}Q_{iLV} = P_{LA} - P_{LV}, \\
L_{oLV}\frac{dQ_{oLV}}{dt} &+ R_{oLV}Q_{oLV} = P_{LV} - P_{A1}, \\
L_A\frac{dQ_A}{dt} &+ R_A Q_A = P_{A1} - P_{A2}, \\
Q_{iRA} &= \frac{P_V - P_{RA}}{R_{iRA}}, \\
Q_{iRV} &= \frac{P_{RA} - P_{RV}}{R_{iRV}}, \\
L_{oRV}\frac{dQ_{oRV}}{dt} + R_{oRV}Q_{oRV} = P_{RV} - P_{APulm}, C_{A1}&\frac{dP_{A1}}{dt} = Q_{oLV} - Q_A, \\
C_{A2}\frac{dP_{A2}}{dt} &= Q_A - \frac{P_{A2} - P_V}{R_{per}} - \frac{P_{A2} - P_C}{R_C}, \\
C_V\frac{dP_V}{dt} &= \frac{P_{A2} - P_V}{R_{per}} - Q_{iRA}, \\
C_C\frac{dP_C}{dt} &= \frac{P_{A2} - P_C}{R_C}, \\
C_{APulm}\frac{dP_{APulm}}{dt} &= Q_{oRV} - \frac{P_{APulm} - P_{VPulm}}{R_{perPulm}}, \\
C_{VPulm}\frac{dP_{VPulm}}{dt} &= Q_{iLA} + \frac{P_{APulm} - P_{VPulm}}{R_{perPulm}}.
\end{aligned}
\tag{A16}
$$

$Q_{iRV}$, $Q_{oRV}$, $Q_{iLV}$, $Q_{oLV}$, $Q_{iRA}$, and $Q_{iLA}$ are blood flows through the tricuspid valve, pulmonary valve, mitral valve, aortic valve, and the flows through systemic and pulmonary veins, respectively; $Q_A$ is the arterial flow. $R$ and $L$ are hydraulic and inertial resistances of the vessels and chamber entrances (subindex $i$) and exits (subindex $o$); $R_{per}$ and $R_{perPulm}$ are the peripheral vascular resistances of systemic and pulmonary circulation systems; $C$ represents the compliances of the vascular reservoirs presented in the model; $R_C$ and $C_C$ characterize the viscoelastic properties of the systemic arteries. Indexes $A1$, $A2$, $V$, $APulm$, and $VPulm$ corresponded to the aorta, large arteries, systemic veins, pulmonary arteries, and veins, respectively.

*A.4. Local Hydraulic Valve Resistances in the Cases of the Valve Pathologies*

The equations for the flows through the aortic and mitral valves in Equation (A16) were modified introducing the quadratic hydraulic valvular resistance and the dependencies of the resistances on orifice area.

$$
\begin{aligned}
L_{**}\frac{S_0}{S}\frac{dQ_{**}}{dt} + \left(R_{**} + R_{sq}|Q_{**}|\right)Q_{**} &= \Delta p, \\
R_{sq} &= C_{sq}\varepsilon_{\text{Re}}\zeta_{sq}S^{-2}, \\
\varepsilon_{\text{Re}} &= \sum_{i=0}^{5} b_i(lg(\text{Re}))^i, \\
\zeta_{sq} &= \left(C_\zeta\left(1 - \frac{S}{S_0}\right)^{0.375} + 1 - \frac{S}{S_0}\right)^2.
\end{aligned}
\tag{A17}
$$

$S$ is the orifice area, and $S_0$ is an area of a completely open valve in healthy conditions; Re is the Reynolds number, $C_\zeta$ and $b_i$ are parameters.

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
