# Peer review of "Hemodynamic Effects of Alpha-Tropomyosin Mutations Associated with Inherited Cardiomyopathies: Multiscale Simulation"

_mathematics, doi:10.3390/math8071169_

Round 1
Reviewer 1 Report
In the manuscript the effects of two cardiomyopathy-associated mutations in regulatory sarcomere protein tropomyosin on heart function are investigated using a multiscale model of the cardiovascular system Moreover, the molecular and cell-level changes in Ca2+ regulation of cardiac muscle caused by these mutations are introduced into the myocardial model of the left ventricle keeping the shape remained the same as in the model of the normal heart, cardiac output and arterial blood pressure reduced. Simulations of left ventricle hypertrophy in the case of the Ile284Val mutation and left ventricle dilatation in the case of the Asp230Asn mutation demonstrated that the left ventricle remodelling partially recovered the stroke volume and arterial blood pressure. It is reported that the end-systolic pressure-volume relation that often used to characterize heart contractility strongly depends on heart geometry and cannot be used as a characteristic of myocardial contractility.
There are few typos that should be removed. For example
On line 133 replace “matched quantitatively data” by “matched the quantitatively data”
On line 135 replace “was” by “and were”
On line 150 replace “reduced compared to” by “reduced as compared to”
On line 201 replace “there are no data” by “there is no data”
On line 203 replace “there are” by “there is”
On line 206 replace “mainly” by “were mainly”
On line 214 replace “data are controversial” by “data is controversial”
On line 458 replace “are available” by “is available”
On line 526 replace “is a the length” by “is the length”
Author Response
We are thankful to the reviewer for the corrections suggested. The following changes were made in the manuscript.
1. On line 133 replace “matched quantitatively data” by “matched the quantitatively data”.
On line 132 “The results of the numerical research matched quantitatively data of clinical guidelines for the classification of the valves pathologies” was rewritten as “The results of the numerical research matched the data of clinical guidelines for the classification of the valves pathologies quantitatively”.
2. On line 135 replace “was” by “and were”.
On line 134 “Numerical methods implemented here are commonly used and have been validated for the convergence of the method was checked by varying the time-step and the size and number of the FEs [16,17]” was rewritten as “Numerical methods implemented here are commonly used and have been validated for the convergence, which was checked by variation of the time-step and the size and number of the FEs [16,17]”.
3. On line 150 replace “reduced compared to” by “reduced as compared to”.
On line 150 “reduced compared to” was replaced by “reduced as compared to” as suggested.
4. On line 201 replace “there are no data” by “there is no data”.
On line 201 “there are no data” was replaced by “there is no data” as suggested.
5. On line 203 replace “there are” by “there is”.
On line 203 “there are clinical research data” was replaced by “there is clinical research data” as suggested.
6. On line 206 replace “mainly” by “were mainly”.
On line 205 “These changes correlated with increased expression of the long N2BA titin isoform in DCM myocardium compared to shorter N2B isoform mainly expressed in normal myocardium” was clarified and rewritten as “These changes correlated with increased expression of the long N2BA titin isoform in DCM myocardium as compared to its expression in normal myocardium, where the shorter N2B isoform was mainly expressed”.
7. On line 214 replace “data are controversial” by “data is controversial”.
On line 215 “data are controversial” was replaced by “data is controversial” as suggested.
8. On line 458 replace “are available” by “is available”.
On line 459 “no 3D data are available” was replaced by “no 3D data is available” as suggested.
9. On line 526 replace “is a the length” by “is the length”.
On line 527 “Wa is a the length of the overlap zone” was replaced by “Wa is the length of the overlap zone” as suggested.
Reviewer 2 Report
This manuscript extends several research publications by the authors on computational modelling of heart function, due to cardiomyopathy-associated mutations. The work is presented at a high standard of clarity and explanation. My only criticism would be on the suitability of Mathematics. How does the experimental data that were used to newly set parameter values in the simulations constitute mathematical research? I am unsure the main achievements were mathematical and wonder whether a biomedical or medical mathematics journal had been considered beforehand.
Author Response
We are grateful to the reviewer for evaluation of our work. Our response to the comment of the reviewer is given below.
1. My only criticism would be on the suitability of Mathematics. How does the experimental data that were used to newly set parameter values in the simulations constitute mathematical research? I am unsure the main achievements were mathematical and wonder whether a biomedical or medical mathematics journal had been considered beforehand.
Although we have modelled the biophysical cell-level processes (the mutations in muscle regulatory proteins), the effects investigated here and the results of the investigation apply more closely to the mechanics of the heart as the solid deformable body than to the cell-level biology/biophysics. Also the mathematical methods were used extensively in our research. For this reason, we suppose that our paper could be at interest for researchers in applied mathematics in the first place. We had considered the publication in biomedical engineering or biophysical journals, however, the special issue of Mathematics “Mathematical Modelling in Biomedicine” (https://www.mdpi.com/journal/mathematics/special_issues/Mathematical_Modelling_Biomedicine) seemed to us being the most suitable place for our publication, as the special issue focuses on interdisciplinary research articles appealing to mathematicians, biologists, and medical workers.
Round 2
Reviewer 2 Report
This seems to me a work of top quality in bio-mathematical modelling based on years of earlier research from the lead authors, and well connected to a broader literature in the field. My only criticism is that the new mathematical methodology could have been explained more directly without use of the intrinsic jargon and biomedical terminology, as requested. At least, that would have given the review a stronger foundation and improved credibility from my perspective.
I had also suggested another reviewer be considered with practical knowledge in deformation tensors or computational cardio-models, but to no avail.